# New Behavioral Handling Test Reveals Temperament Differences in Native Japanese Chickens

**DOI:** 10.3390/ani13223556

**Published:** 2023-11-17

**Authors:** Akira Ishikawa, Tomoka Takanuma, Norikazu Hashimoto, Tatsuhiko Goto, Masaoki Tsudzuki

**Affiliations:** 1Laboratory of Animal Genetics and Breeding, Graduate School of Bioagricultural Sciences, Nagoya University, Nagoya 464-8601, Japan; 2Laboratory of Poultry, Livestock Experiment Station, Wakayama Prefecture, Hidaka-Gun, Wakayama 644-1111, Japan; hashimoto_n0027@pref.wakayama.lg.jp; 3Research Center for Global Agromedicine, Obihiro University of Agriculture and Veterinary Medicine, Obihiro 080-8555, Japan; tats.goto@obihiro.ac.jp; 4Laboratory of Animal Breeding and Genetics, Graduate School of Integrated Sciences for Life, Hiroshima University, Higashi-Hiroshima 739-8525, Japan; tsudzuki@hiroshima-u.ac.jp

**Keywords:** behavioral test, fear, native Japanese chicken, stress, temperament

## Abstract

**Simple Summary:**

The poultry industry recognizes that handling day-old chicks in commercial hatcheries can lead to long-lasting changes in their behavior. However, these hatchery-related stresses are more intense and complex than those measured by traditional behavioral tests. This study developed a new behavioral handling test for day-old chickens by incorporating concepts from behavioral tests used with both young and adult birds. In the new test, 96 two-day-old chicks from seven breeds of native Japanese and Western chickens were used to evaluate 10 behavioral traits, including vocalization frequency and responses to human interaction. The results classified chicken temperaments into three categories: bustle, aggression, and timidity. These categories were used to classify the seven breeds. This new handling test provides a valuable tool for understanding the behavior of various chicken breeds and offers insights into their complex behaviors.

**Abstract:**

It is well known in the poultry industry that fear and stress experienced during the handling of day-old chicks in commercial hatcheries can have long-lasting effects on their behavior later in life. These hatchery-related stresses are more intense and complex than those encountered in traditional behavioral tests. Consequently, a single behavioral test may not be sufficient to measure hatchery stresses and chicken temperament. In this study, we developed a new behavioral handling test for day-old chickens, which incorporated concepts from established behavioral tests used with both young and adult birds. The new test assessed 10 behavioral traits, including vocalization frequency and responses to human interaction. It was conducted on 96 two-day-old chicks from seven breeds of native Japanese and Western chickens. The results of the principal component analysis classified chicken temperaments into three distinct categories: bustle, aggression, and timidity. Using these categories, the seven breeds were classified into five groups, each with distinct temperaments. This study highlights the reliability and value of the new handling test in characterizing the temperaments of various chicken breeds and provides insights into the complex behaviors of chickens.

## 1. Introduction

Prenatal and postnatal stresses have been established to exert both short- and long-term effects on the brain, behavior, and cognition of animals [1]. In the commercial poultry industry, day-old chicks are exposed to a multitude of stresses during pre-hatching and post-hatching hatchery handling, which includes processes like incubation, sex sorting, vaccination, and transportation [2,3]. These hatchery-related stresses are known to have lasting impacts on the subsequent life of chickens. For example, one-day-old laying hens that experienced noisy incubation and human handling at hatching exhibited lower body weights compared to control hens and displayed more active behavior in a novel arena test [3]. Twenty-week-old chickens exposed to stressful hatchery procedures at hatching exhibited higher levels of feather pecking damage and injuries on their combs and wattles compared to control chickens [2]. Furthermore, 40-day-old laying hens that experienced fear in the tonic immobility test at 7 days of age showed a moderate genetic correlation with feather pecking and aggressive pecking [4]. Additionally, the Nagoya, a native Japanese breed known for its high-quality meat and egg production, is observed to have a timid temperament [5]. Chicken flocks of this breed often exhibit startled reactions to unexpected noises and other environmental stimuli, tending to huddle together, which can result in crush deaths [6]. The manifestation of abnormal behaviors due to stress and fear can compromise animal welfare and lead to economic losses. Therefore, developing genetically docile chickens that are less susceptible to perinatal stress and fear would be advantageous to the chicken industry.

Several traditional behavioral tests are available to evaluate fear responses in poultry, including well-known tests like the tonic immobility test, the open field test, and the novel object test [7]. The tonic immobility test assesses the fear level of a bird by inducing strong fear through manual restraint [7,8,9,10]. The open field test evaluates general fear responses and social reinstatement effects by exposing a bird to mild fear in a novel environment [7,11]. The novel object test assesses the general fear responses of a bird when approached by a novel object or a human [7,12,13]. These traditional tests have been conducted on young and adult chickens [7,8,9,10,11,12,13], but they were not typically used with day-old chickens until our research group introduced them. Recently, a handling test was developed to assess the level of tameness in adult chickens [14]. This test involves close human hand approaches or direct contact with a bird to evoke fear or stress responses. However, such a handling test has not been reported for day-old chickens. Day-old chicks typically encounter fear and stress stimuli during commercial hatchery handling, which are very likely to be more intense and complex than those addressed in traditional behavioral tests. Consequently, it becomes challenging to assess their fear responses using a single behavioral test. Furthermore, relying on only one behavioral test to evaluate the temperament of chickens has its limitations. Previous studies, such as [15,16,17], have addressed these challenges by employing multiple behavioral tests simultaneously. However, this approach requires labor-intensive experimental designs. To overcome these challenges and limitations, it is highly desirable to develop a new, single behavioral test that can simultaneously assess multiple aspects of behavioral temperament in chickens and that is easy to implement.

In this study, we developed a new behavioral handling test for day-old chickens. This new test incorporated concepts from the traditional novel arena and novel object tests [7,8,9,10,11,12,13] as well as insights from the handling test [14], which are traditionally applied to both young and adult chickens. Furthermore, we also assessed the effectiveness of this new handling test in characterizing the complex behavioral temperaments of day-old chickens from various native Japanese chicken breeds and a distinct Western chicken breed. It is important to note that the handling test was not designed to replicate the intense fear and stress experienced in commercial hatcheries. Instead, its purpose was to reveal and characterize the natural temperament and behavioral responses of the day-old chicks.

## 2. Materials and Methods

### 2.1. Ethical Note

All animal experiments were conducted in accordance with the guidelines for the care and use of laboratory animals at Nagoya University, Japan. The protocols were approved by the Nagoya University Animal Research Committee (approval no. AGR2019016).

### 2.2. Animals

A total of 96 two-day-old chickens were used. The research involved five distinct native Japanese chicken breeds: Chabo (CHB), Ingie (IG), Oh-Shamo (OSM), Ryujin-Jidori (RYU), and Tosa-Kukin (TKU). Within the CHB breed, two separate lines were derived from Hiroshima University (H) and Nagoya University (N). Furthermore, a Western breed was represented by PNP, a highly inbred line derived from the Fayoumi breed. Detailed information on these six breeds and lines is given in Table 1. Fertilized eggs for all breeds were obtained from the resource centers listed in Table 1. The eggs were hatched in the chicken house of Nagoya University, and the hatched chicks were handled according to previously described methods [15]. Briefly, the eggs from all breeds were placed in an incubator at 9:00 AM. Approximately 18 days after incubation, egg candling was performed to monitor the hatching progress. Hatching was confirmed the evening before the scheduled hatching date (21 days after incubation). Only chicks that hatched on the morning of the scheduled hatching day were selected and continuously reared in a small brooder at 32 °C. Before the behavioral test, the chicks were given only water.

### 2.3. Handling Test

Two-day-old chicks that were verified to be free of deformities and capable of standing were used in the new handling test. The test was conducted by placing the chicks in an arena measuring 40 cm in length, 21 cm in width, and 29 cm in height, as shown in Figure 1. The arena was enclosed on three sides with wire mesh. The floor of the arena was divided into six equal fields, marked by lines on the floor and covered with a plastic board, to facilitate the measurement of the bird’s movement distance during testing.

The protocols of the handling test are summarized in Figure 2. First, each bird was placed in field 1 of the arena and covered with a cylindrical steel can (height 12.5 cm, diameter 10 cm) for 15 s to keep it in darkness. The can was then opened to begin the handling test. Using a human hand and a homemade cotton swab, the bird was exposed to three different stimuli: human proximity, cotton swab contact, and human handling. The behavior of the bird was recorded for 7 min using a video camera (Handycam HDR-PJ675, SONY, Tokyo, Japan). The records were analyzed according to the criteria for 10 behavioral traits in the handling test, as shown in Table 2. Distress vocalizations, defined as repetitive, high-energy, and relatively loud calls [21], were distinguished from general vocalizations in this study. These 10 behavioral traits were classified into three temperament categories: traits related to bustle, aggression, and timidity. For each trait, the number of occurrences was counted. After the handling test, the body weight of the bird was recorded.

### 2.4. Sexing

After conducting the handling test, blood was collected from the carotid artery of each chick using a heparin-containing microtube. The plasma and precipitated red blood cells were separated by centrifugation at 3500 rpm for 10 min. Genomic DNA was extracted from the red blood cells. Sex determination was performed by PCR amplification of the chromo-helicase-DNA binding protein (*CHD*) gene located on Z and W chromosomes, following previously described methods [22].

### 2.5. Statistical Analysis

Statistical analyses of the data for all behavioral traits were performed using the software package JMP Pro version 16.2.0 (SAS Institute Japan, Tokyo, Japan). Prior to breed comparisons, raw trait data were tested with a linear regression model of JMP Pro to determine whether they were influenced by sex and body weight. Only body weight showed significance at a nominal *p* < 0.05 (Appendix A), and it was used to adjust the raw data for four traits (Appendix A). The average body weight for each breed is presented in Appendix A. The residuals obtained after adjustments were utilized for subsequent statistical analyses. Differences in each trait between breeds were assessed using the Kruskal–Wallis test, followed by the Steel–Dwass post hoc test. The *p* values for the Kruskal–Wallis test were obtained by the chi-square values for the one-way test. To understand the behavioral characteristics of each breed, principal component analysis using a correlation matrix was performed on the adjusted trait data.

## 3. Results

### Handling Test

Initially, we assessed 19 handling traits, measuring six traits related to bustle, 10 traits related to aggression, and three traits related to timidity. Subsequently, a principal component analysis of these 19 traits was conducted. The resulting factor loading values for the first and second principal component axes are shown in Appendix A. Traits with low factor loading values (indicating a low contribution to the principal component axis) or those with similar directions for the factor loading vectors were either excluded from the analysis or combined to create a new single trait. For example, the trait for the number of instances of approaching the hand on its own was excluded due to its low contributions to both principal component axes, while traits related to floor/cage pecking were merged. The factor loading value of the first principal component axis for the number of times the cage was pecked was higher than that for the number of times the floor was pecked, where exploratory behavior might be included. However, it was nearly the same value for the number of times the floor or cage was pecked, which integrated both floor and cage pecking instances (Appendix A). This iterative process was continued until an optimal set of traits, i.e., 10 traits, was achieved.

The means of 10 handling traits were compared among the seven breeds, and the results are summarized in Appendix A. The statistical analyses, using the Kruskal–Wallis test and the Steel–Dwass post hoc test, revealed that only one trait (escaping stimulus) showed no significant differences at *p* < 0.05 among the seven breeds, whereas the rest of the nine traits were all found to be significant. Among the seven breeds, PNP displayed unique characteristics by having the highest and lowest occurrences for six traits. The three traits with the highest occurrences were distress vocalization, moving, and escaping the cage, while the three traits with the lowest occurrences were general vocalization, sleeping, and approaching the wall. TKU showed the highest values for two traits (sleeping and floor/cage pecking) and the lowest values for three traits (moving, escaping cage, and biting), indicating that this breed might have a relatively docile nature. In contrast, the OSM breed exhibited the highest value for only one trait, biting, indicating the most aggressive disposition. On the other hand, RYU did not show the highest or lowest values for any of the traits.

To comprehensively assess behavioral characteristics, principal component analysis was performed on the nine handling traits (excluding escaping stimulus) in the seven breeds. The results are summarized in Figure 3, and details such as factor loading values and percentages of variance contributed to the first and second principal component axes are shown in Appendix A. The first and second principal component axes accounted for 32.9% and 16.6% of the total variance, respectively. For the first principal component axis, six of the nine traits had positive factor loadings, with distress vocalization and moving showing nearly the highest loading values, while three traits had negative factor loadings, with general vocalization and sleeping showing nearly the lowest loading values (Appendix A). Notably, the four traits with the highest and lowest loading values (distress vocalization, moving, general vocalization, and sleeping) were all associated with the bustling nature of handling behavior, as seen in Table 2. This suggested that the first principal component axis predominantly explained the bustling aspects of chick behavior. Chicks with more positive first principal component scores tended to exhibit more bustling behavior. The score plot patterns (Figure 3A) indicated that OSM and PNP breeds were the most bustling. For the second principal component axis, five traits had positive factor loadings, with biting having the highest loading value, while four traits had negative factor loadings, with a surprised voice having the lowest loading value (Appendix A). Biting was associated with aggression, while a surprised voice was linked to timidity, as shown in Table 2. Consequently, the second principal component axis clearly explained the aggression and timidity of chick temperament. The OSM breed, having the highest biting value (Appendix A), appeared to be the most aggressive (Figure 3A).

After excluding the OSM and PNP breeds that exhibited extreme temperament in the handling test, the remaining five native Japanese breeds were again subjected to principal component analysis to further investigate their temperament characteristics. The results are shown in Figure 4, and details such as factor loading values and percentages of variance contributed to the first and second principal component axes are shown in Appendix A. The first and second principal component axes accounted for 32.6% and 16.7% of the total variance, respectively. Consistent with the findings from the initial principal component analysis that included all seven breeds, distress vocalization and moving had nearly the highest loading values, while sleeping had the lowest loading value. This reaffirmed that the first principal component axis predominantly described the bustling nature of behavior. On the other hand, for the second principal component axis, general vocalization displayed the highest positive loading value, whereas floor/cage pecking had the lowest negative loading value. This suggested that this axis encapsulated a combination of bustling and timidity. The score plot patterns (Figure 4A) separated the CHB-N breed from the other four breeds for two traits (general vocalization and floor/cage pecking). The uniqueness of CHB-N was supported by the significant differences in mean scores for the first and second principal component axes between this breed and other breeds (Figure 4C,D). Furthermore, the remaining four breeds could be roughly classified into two groups: the IG and TKU breeds, and the CHB-H and RYU breeds.

## 4. Discussion

In this study, we successfully developed a new handling test specifically designed for day-old chickens by integrating concepts from the novel arena test and the novel object test and insights from the adult handling test [7,14]. The varying degrees of fear and stress evoked by this new test enabled us to classify chicken temperament into three categories: bustle, aggression, and timidity. Using these three temperaments, we characterized the seven native Japanese and Western chicken breeds into five groups, each with distinct temperaments, although further research with a larger number of individuals per breed is necessary.

The OSM breed was uniquely characterized by having the most aggressive temperament among the breeds examined. OSM, known as a Japanese game breed for cockfighting, is recognized for its robust body and aggressive nature [5]. Remarkably, despite using day-old chicks, our handling test effectively revealed the aggressive nature of OSM. Another interesting breed is PNP, a highly inbred line derived from three roosters and two hens of the Fayoumi breed, which has been maintained as a closed breeding colony with several pairs per generation [18]. PNP exhibited more exploratory behavior and higher locomotor activity compared to other breeds, indicating a bustling nature.

Furthermore, we observed variations in handling traits between the two lines of the CHB breed. These variations may reflect underlying genetic differences between the two CHB lines, which could be attributed to their distinct ancestorial base populations. According to the literature, the N line of CHB was established using founder chickens obtained from a chicken fancier in 2013 [18]. On the other hand, the H line of CHB was developed from founder chickens sourced from chicken fanciers in Hiroshima Prefecture, Japan (Tsudzuki, personal communication [5]). Additional research and analyses would be necessary to further clarify the genetic and behavioral differences between these two CHB lines.

The handling test also successfully distinguished two groups: the CHB-H and RYU breeds, which exhibited less docile behavior, and the IG and TKU breeds, which displayed a more docile nature. However, based on a phylogenic tree based on the polymorphisms of 27 microsatellite DNA markers, the RYU, IG, and TKU breeds each belong to three different clades, whereas the CHB and TKU breeds are grouped together in one clade [23]. It is important to note that only one TKU chicken was used in this phylogenic study [23]. However, another phylogenic tree based on the polymorphisms of 28 microsatellite DNA markers indicates separate clades for the CHB and TKU breeds [24]. Variations in sample sources for both breeds likely account for these inconsistencies in genetic relationships. Therefore, it is highly probable that the temperaments of the four breeds reflect their respective breeding histories and origins.

To the best of our knowledge, this study is the first behavioral investigation of the CHB, RYU, and PNP breeds. In contrast, there have been a few previous studies using day-old chicks or young chicks of the IG, OSM, and TKU breeds. In our previous study [16], the tonic immobility test and the open field test were conducted with day-old chickens from six native Japanese chicken breeds (IG, OSM, TKU, Nagoya (NAG), Tosa-Jidori, and Ukokkei) and two White Leghorn (WL) lines. The results revealed intriguing variations in behavioral temperaments among the breeds. Notably, OSM exhibited the least sensitivity to innate fear in both the tonic immobility and open field tests, while IG and TKU showed moderate sensitivity to open-field fear. These previous findings align with the results of the handling test in the present study. In a previous report by another research group, three behavioral tests (the tonic immobility test, the isolation test, and the manual restrain test) were conducted together to assess fearfulness in young chicks of the TKU breed and the native Japanese Yakido breed [17]. This report indicated that TKU exhibited docile behavior, while Yakido displayed aggression. In our present study, we observed that OSM displayed an aggressive temperament, whereas both IG and TKU exhibited similar docile temperaments. This consistency of these outcomes across various tests and studies further reinforces the findings from the handling test for these chicken breeds.

It is widely accepted that behavior is a complex trait influenced by multiple genetic loci, known as quantitative trait loci (QTLs), environmental factors, and their interactions. A large number of QTLs affecting chicken behavior have been reported, most of which are cataloged in the Chicken QTL Database [25]. In a previous QTL mapping study using an F2 day-old chicken population from a cross between the NAG and WL breeds, we identified four QTLs associated with fear responses in the tonic immobility test on chromosomes 1–3 and 24 and three QTLs associated with fear responses in the open field test on chromosomes 2, 4, and 7 [26]. Notably, the neuropeptide Y receptor Y5 (*NPY5R*) and *LOC101749214* were pinpointed as candidate genes for the primary open field QTL on chromosome 4 [27]. Furthermore, in another F2 population of day-old chickens obtained from a cross between the OSM and WL breeds, we identified two QTLs for fear responses in the open field test on chromosomes 1 and 2 [28]. Importantly, the nine QTLs for the tonic immobility and open field tests identified in these two populations are all located in different genomic regions [26,28]. This indicates that innate fear responses in the tonic immobility and open field tests are under distinct genetic controls and exhibit variations among breeds. Consequently, the fear responses observed in the handling test likely have a distinct genetic basis compared to those in the open field and tonic immobility tests.

## 5. Conclusions

This study successfully developed a new handling test for day-old chickens by incorporating the concepts of the novel arena test, novel object test, and handling test, which are behavioral tests traditionally used on young and adult chickens. The results of this new handling test proved to be a valuable and sensitive tool for characterizing the temperament of various chicken breeds through a single behavioral test.

## Figures and Tables

**Figure 1 animals-13-03556-f001:**
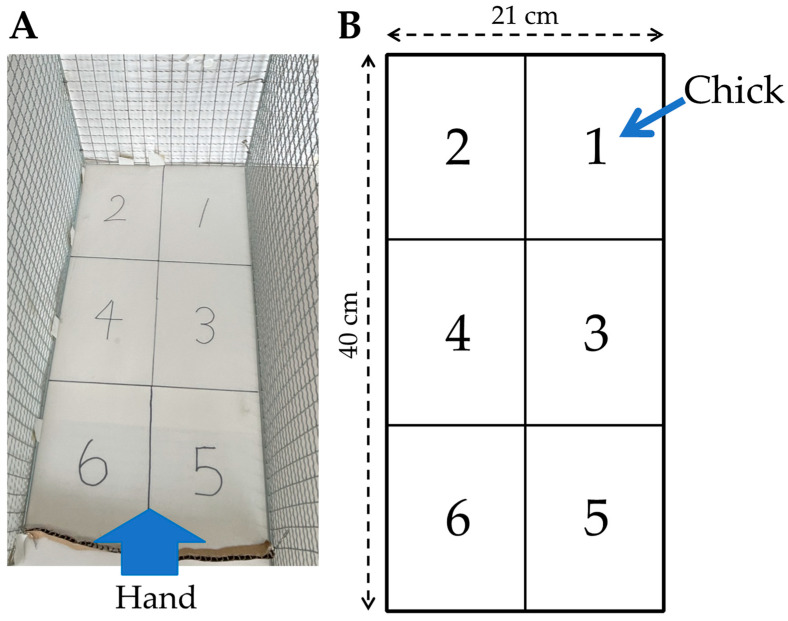
Arena used for the new handling test developed in this study. (**A**) Photograph of the arena; (**B**) Illustration of the arena. The bold arrow indicates the position from which a human hand approaches a bird. The thin arrow indicates the position on which the chick is placed at the beginning of the test.

**Figure 2 animals-13-03556-f002:**
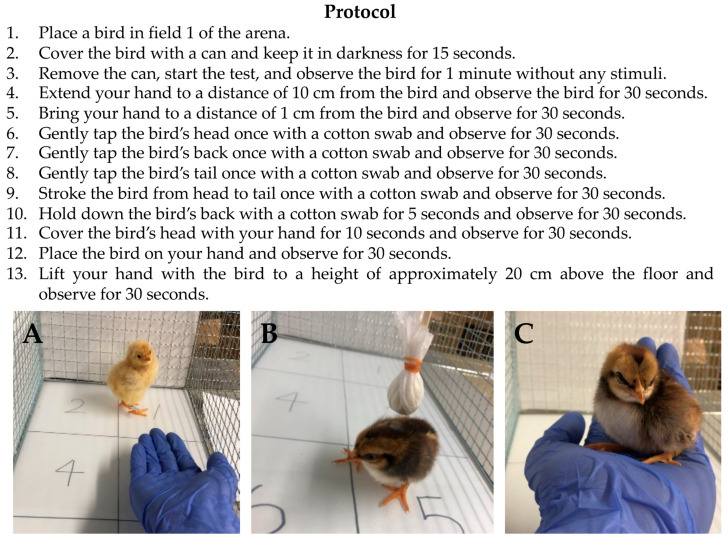
Experiment protocols for the new handling test developed in this study, showing (**A**) Protocol 4, (**B**) Protocol 7, and (**C**) Protocol 13. Using a human hand and a cotton swab, three different stimuli (human proximity, cotton swab contact, and human handling) are applied to a bird, and the behavior of the bird was recorded on video for 7 min.

**Figure 3 animals-13-03556-f003:**
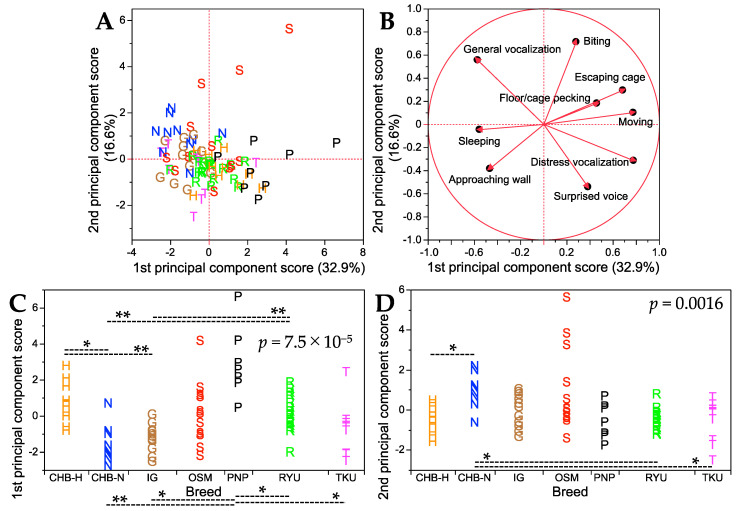
Summary of principal component analysis results for nine handling traits in seven breeds. (**A**) Score plot for the first and second principal component axes. The percentage of total variance explained by each principal component axis is shown in parentheses. (**B**) Factor loading plot, where each trait is represented by a vector. See Appendix A for detailed factor loading values and other parameters on the first and second principal component axes. (**C**) Breed comparisons for the first principal component scores. (**D**) Breed comparisons for the second principal component scores. *p* values in (**C**,**D**) were obtained from the Kruskal–Wallis test for each of the first and second principal component scores among the seven breeds. Dashed lines indicate significant differences in the first and second principal component scores between the two breeds at *p* < 0.05 (*) and *p* < 0.01 (**) (Steel–Dwass post hoc test). Single letters in (**A**–**D**) represent breed abbreviations and each letter denotes an individual of the breed.

**Figure 4 animals-13-03556-f004:**
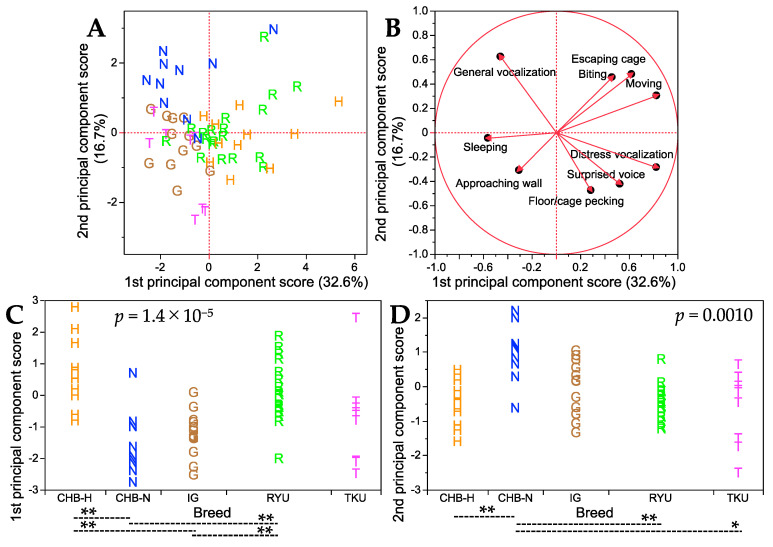
Summary of principal component analysis results for nine handling traits in five breeds. (**A**) Score plot for the first and second principal component axes. The percentage of total variance explained by each principal component axis is shown in parentheses. (**B**) Factor loading plot, where each trait is represented by a vector. See Appendix A for detailed factor loading values and other parameters on the first and second principal component axes. (**C**) Breed comparisons for the first principal component scores. (**D**) Breed comparisons for the second principal component scores. *p* values in (**C**,**D**) were obtained from Kruskal–Wallis test for each of the first and second principal component scores among seven breeds. Dashed lines indicate significant differences in the first and second principal component scores between the two breeds at *p* < 0.05 (*) and *p* < 0.01 (**) (Steel–Dwass post hoc test). Single letters in (**A**–**D**) represent breed abbreviations and each letter denotes an individual of the breed.

**Table 1 animals-13-03556-t001:** List of seven breeds of native Japanese and Western chickens used in this study.

		Number of Animals		Source and Origin of Breed ^1^
Breed or Line	Abbreviation	Male	Female	
Chabo H line	CHB-H	8	3	JABPC
Chabo N line	CHB -N	5	5	ABRC [18]
Ingie	IG	4	8	JABPC [19]
Ryujin-Jidori	RYU	11	13	LPLESWP [20]
Oh-Shamo	OSM	7	5	JABPC [5]
Tosa-Kukin	TKU	2	7	JABPC [5]
PNP	PNP	14	4	ABRC [18]

^1^ ABRC, Avian Bioscience Research Center, Graduate School of Bioagricultural Sciences, Nagoya University, Japan; JABPC, Japanese Avian Bioresource Project Research Center, Hiroshima University, Japan; LPLESWP, Laboratory of Poultry, Livestock Experimental Station, Wakayama Prefecture, Japan. Numbers in square brackets indicate the literature number of origin of each breed or line; CHB-H was derived from founder chickens obtained from chicken fanciers in Hiroshima Prefecture, Japan (Tsudzuki, personal communication [5]).

**Table 2 animals-13-03556-t002:** Criteria for 10 behavioral traits used in the new handling test developed.

Trait	Abbreviation	Criterion
Traits related to bustle		
Number of distress vocalizations	Distress vocalization	More vocalizations indicate more clamor
Number of movements across the line	Moving	Number of times the bird crossed the line during the 30 s observation period after stimulation; more moves indicate more noise
Number of escape attempts from the cage	Escaping cage	More attempts indicate more clamor
Number of general vocalizations other than distress vocalizations	General vocalization	More vocalizations indicate relatively fewer distress vocalizations and less noise
Number of sleeps	Sleeping	Number of sleeps in each stimulus phase; more sleep indicates less bustle
Traits related to aggression		
Number of times the floor or cage was pecked	Floor/cage pecking	More pecks indicate more aggression
Number of bites	Biting	More bites indicate more aggression
Traits related to timidity		
Number of voices raised or surprised when stimulated	Surprised voice	More voices indicate more timidity
Number of escape attempts when stimulated	Escaping stimulus	More attempts indicate more timidity
Number of times approaching the cage wall	Approaching wall	More approaches indicate more timidity

## Data Availability

The data present in this study are available in Appendix A.

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
