# Peer review of "New Behavioral Handling Test Reveals Temperament Differences in Native Japanese Chickens"

_animals, 2023, doi:10.3390/ani13223556_

Round 1

Reviewer 1 Report

Comments and Suggestions for Authors

This is a nice study using a newly developed behavooral test to address the long-term effects of stress and fear in newly hatched chicks in commercial hatcheries. The experiences chicks undergo during handling in these hatcheries can have substantial impacts on their behavior later in life. So, to tackle this issue, the authors have developed an innovative behavioral test for two-day-old chicks, using established concepts from behavioral tests. The combination of several behavioral tests in just one test and the easy feasibility make this a promising new approach to assess a chicken's temperament.

The manuscript is written in a clear and well-structured manner, especially the results are clearly presented, but there are also some points that should be improved.
Starting with the introduction. I was wondering if there is already literature that more docile chickens are indeed less susceptible to perinatal stresses as the character of each chicken not only depends on its genetic predispositions and experiences but also strengthens with age. Further, I am missing some information on which behavioral tests besides TI and OF are already used to assess chickens' temperaments (e.g. you mention the novel object test at the end of your introduction but never before), if they are reliable or can even be done with chicks and if they produce the same results when tested with chicks or adults. What was your hypothesis?

The methods part is well structured and reproducibility is given, but there are some things that are unclear. On what (literature) basis did you classify the 10 behavioral traits into the 3 temperament categories? Why did you assign cage/floor pecks to aggression, isn't it rather an exploratory behavior? How did you classify the vocalizations of the chicks as distress calls or surprised voices? Did several observers classify the voalizations and if yes, what about the interobserver-reliability?

The results part is nicely presented and clearly understandable. Figures and tables are appropriate and show the data properly.

Discussion&Conclusions: I'm missing more information about what other studies found about the breeds you used in your study, indepenedent of your own studies. Are your results in line with their results? Do you plan to apply your test to the same chickens at different ages (as chick and as adult) to validate its reliability? Coming to your original question of the influence of handling and stressors at hatching on the chicken's behavior later on in life - were your handling methods after hatch comparable to those at commercial hatcheries? Or were your handling methods as stress-free as possible, aiming to capture initially the most neutral picture of the chicks' temperaments and to validate your test method then in further experiments with commercial hatchery stressors?

Additionally oyu should check your manuscript carefully for typs (e.g. Line 255 "outcom").

Overall, this study seems to provide a reliable method for characterizing various chicken temperaments but also offers insight into the complex behavior of these animals. These insights could potentially contribute to enhancing the well-being and quality of life of chickens by considering the test results in breeding and enabling hatcheries to adjust and design environments accordingly.

Author Response

Response to Reviewer 1

Starting with the introduction. I was wondering if there is already literature that more docile chickens are indeed less susceptible to perinatal stresses as the character of each chicken not only depends on its genetic predispositions and experiences but also strengthens with age. Further, I am missing some information on which behavioral tests besides TI and OF are already used to assess chickens' temperaments (e.g. you mention the novel object test at the end of your introduction but never before), if they are reliable or can even be done with chicks and if they produce the same results when tested with chicks or adults. What was your hypothesis?

Response: Currently, there are no reports indicating that docile chickens are less susceptible to perinatal stress, although it is highly likely. A new object test was added to the Introduction. As of now, there are no reports of this test being conducted on both chicks and adult chickens. These areas of research present intriguing opportunities for future investigations.

The methods part is well structured and reproducibility is given, but there are some things that are unclear. On what (literature) basis did you classify the 10 behavioral traits into the 3 temperament categories? Why did you assign cage/floor pecks to aggression, isn't it rather an exploratory behavior? How did you classify the vocalizations of the chicks as distress calls or surprised voices? Did several observers classify the voalizations and if yes, what about the interobserver-reliability?

Response: We did not adhere to any specific literature in determining the 10 traits. Initially, we assessed more than 10 behavioral traits. Specifically, we measured six traits related to bustle, 10 traits related to aggression, and three traits related to timidity. Subsequently, we conducted a principal component analysis on these 19 traits. If any of the traits showed low values for their factor loadings (i.e., low contribution to the principal component axis) or had approximately the same directions for their factor loading vectors, they were either excluded from the study or combined to create a new single trait. This iterative process was repeated until an optimal set of traits, i.e., 10 traits, was obtained. This explanation was added to the Results.

The cage/floor pecks trait was initially divided into three traits: cage pecks, floor pecks, and cage/floor pecks. As noted by the reviewer, the floor pecks trait may include exploratory behavior. However, the factor loading value of cage pecks (0.61), a clear indicator of aggression, was nearly identical to the factor loading value of cage/floor pecks (0.64), but greater than that of floor pecks (0.43). Thus, the cage/floor pecks trait is ultimately selected and included in the aggressive traits.

The difference between a distress call and a surprised voice is distinguished by the difference in tone and intensity of the voice. Two persons were able to easily distinguish between these two types of vocalizations.

Discussion&Conclusions: I'm missing more information about what other studies found about the breeds you used in your study, indepenedent of your own studies. Are your results in line with their results?

Response: To the best of our knowledge, this study is the first behavioral study on the CHB, RYU, and PNP breeds. For the IG, OSM, and TKU breeds, there are a few previous reports on behavioral studies. Another reference was cited, and some discussion was added in Discussion.

Do you plan to apply your test to the same chickens at different ages (as chick and as adult) to validate its reliability?

Response: Thank you for the interesting experimental idea. We have no plan for such a study at this time, but it would be beneficial to do so in the near future.

Coming to your original question of the influence of handling and stressors at hatching on the chicken's behavior later on in life - were your handling methods after hatch comparable to those at commercial hatcheries? Or were your handling methods as stress-free as possible, aiming to capture initially the most neutral picture of the chicks' temperaments and to validate your test method then in further experiments with commercial hatchery stressors?

Response: Since stress and fear in commercial hatcheries are likely to be greater than in handling tests, this study provides the most neutral view of chick temperament. The purpose of this study was to determine the differences in temperament among day-old chicks in different Japanese breeds. We added this notice to the last paragraph of the Introduction. In addition, the temperamental differences are very likely to link to genetic differences, as discussed in the last paragraph of the Discussion.

Additionally oyu should check your manuscript carefully for typs (e.g. Line 255 "outcom").

Response: This spelling error was corrected.

Reviewer 2 Report

Comments and Suggestions for Authors

Introduction- you should clearly explain the objective of assessing a new behavioural test for the chicks rather than using the existing conventional behaviour tests.

Methodology- Better to include the hatchery conditions of all breeds.

What is can?....Describe the can you used in the test

Please check the protocol for correct wordings (I.e. Birds' tail)

Statistical analysis- you talk about body weight....but there are no results related to the body weight of chicks. Please give the average Body weight of each chick breed in the methodology if you are including any results related to body weight.

better to replace the word environmental factors for sex and body weight

Results- Please clearly define PC1 and PC2

Better to avoid too much of abbreviations - difficult to read 

Line 175- what is table S4?

Discussion

what do you mean by two groups in Line 237

Conclusion

Better to write the conclusion by considering the objective of the study. You better not to generalize the results for chickens. I suggest ti delete the last sentence in the conclusion.

Comments on the Quality of English Language

Need to carefully check the words for the correct spelling.

Minor revisions of a few sentences are needed

Author Response

Response to Reviewer 2

Introduction- you should clearly explain the objective of assessing a new behavioural test for the chicks rather than using the existing conventional behaviour tests.

Response: Some sentences in the Introduction were revised according to the advice provided.

Methodology- Better to include the hatchery conditions of all breeds.

Response: The hatchery conditions were added to the text, and they were the same for all breeds.

What is can?....Describe the can you used in the test

Response: The can is a cylindrical steel can (height 12.5 cm, diameter 10 cm) for confectionery use. it was thoroughly washed and dried before use.

Please check the protocol for correct wordings (I.e. Birds' tail)

Response: The wording was checked and corrected.

Statistical analysis- you talk about body weight....but there are no results related to the body weight of chicks. Please give the average Body weight of each chick breed in the methodology if you are including any results related to body weight.

Response: The average body weights and SEs for all breeds were added in Table S3. The title of Table S3 was changed.

better to replace the word environmental factors for sex and body weight

Response: Replaced with sex and body weight. To reflect this replacement, a related sentence was revised, and the title of Table S1 was changed.

Results- Please clearly define PC1 and PC2

Response: The results of the principal component analysis were rephrased to clarify the definitions of the first and second principal component axes.

Better to avoid too much of abbreviations - difficult to read

Response: The abbreviations “PC”, “TI”, and “OF” were removed and spelled out.

Line 175- what is table S4?

Response: Added brief explanations of Table S4 and Table S5 in the text.

Discussion

what do you mean by two groups in Line 237

Response: One group consists of CHB-H and RYU breeds. The other group consists of IG and TKU breeds. The text was rephrased for clarity.

Conclusion

Better to write the conclusion by considering the objective of the study. You better not to generalize the results for chickens. I suggest ti delete the last sentence in the conclusion.

Response: The last sentence was deleted, and the conclusion was revised following the advice provided.

 Comments on the Quality of English Language

Need to carefully check the words for the correct spelling.

Response: The wording was carefully checked, and spelling errors were corrected.

Minor revisions of a few sentences are needed

Response: The sentences were revised for clarity and readability.

Round 2

Reviewer 1 Report

Comments and Suggestions for Authors

Thank you for addressing my concerns in your revised manuscript. I still think it would be helpful and increase transparency if you would provide an additional table or something where you list the exact 19 traits you measured intitially, possibly also with their factor loadings, and a brief description how you measured them. E.g., for the different calls how they were specifically characterized, you should consider that someone else should easily be able to replicate your study.

Author Response

Response to Reviewer 1

Thank you for addressing my concerns in your revised manuscript. I still think it would be helpful and increase transparency if you would provide an additional table or something where you list the exact 19 traits you measured intitially, possibly also with their factor loadings, and a brief description how you measured them. E.g., for the different calls how they were specifically characterized, you should consider that someone else should easily be able to replicate your study.

Response: We generated a new Table S4 displaying the factor loading values for the initial 19 handling traits along with their concise trait explanations. In our initial response, we overlooked including a reference that defines distress vocalizations as repetitive, high-energy, and relatively loud calls (Herborn et al., 2020, J. R. Soc. Interface 17, 20200086). In the context of this study, vocalizations other than distress vocalizations are referred to as general vocalizations. These and related explanations were added to the Materials & Methods and Results.